# A Systematic Review of the Role of Purinergic Signalling Pathway in the Treatment of COVID-19

**DOI:** 10.3390/ijms24097865

**Published:** 2023-04-26

**Authors:** Vitoria Guero Korb, Iago Carvalho Schultz, Liziane Raquel Beckenkamp, Márcia Rosângela Wink

**Affiliations:** 1Laboratório de Biologia Celular, Universidade Federal de Ciências da Saúde de Porto Alegre (UFCSPA), Porto Alegre 90050-170, RS, Brazil; 2Departamento de Ciências Básicas da Saúde, Universidade Federal de Ciências da Saúde de Porto Alegre (UFCSPA), Rua Sarmento Leite, 245, Sala 304 Centro, Porto Alegre 90050-170, RS, Brazil

**Keywords:** COVID-19, SARS-CoV-2, purinergic signalling, purinergic receptors

## Abstract

The coronavirus disease 2019 (COVID-19) pandemic, caused by severe acute respiratory syndrome coronavirus 2 (SARS-CoV-2), has become a global health concern. Three years since its origin, despite the approval of vaccines and specific treatments against this new coronavirus, there are still high rates of infection, hospitalization, and mortality in some countries. COVID-19 is characterised by a high inflammatory state and coagulation disturbances that may be linked to purinergic signalling molecules such as adenosine triphosphate (ATP), adenosine diphosphate (ADP), adenosine (ADO), and purinergic receptors (P1 and P2). These nucleotides/nucleosides play important roles in cellular processes, such as immunomodulation, blood clot formation, and vasodilation, which are affected during SARS-CoV-2 infection. Therefore, drugs targeting this purinergic pathway, currently used for other pathologies, are being evaluated in preclinical and clinical trials for COVID-19. In this review, we focus on the potential of these drugs to control the release, degradation, and reuptake of these extracellular nucleotides and nucleosides to treat COVID-19. Drugs targeting the P1 receptors could have therapeutic efficacy due to their capacity to modulate the cytokine storm and the immune response. Those acting in P2X7, which is linked to NLRP3 inflammasome activation, are also valuable candidates as they can reduce the release of pro-inflammatory cytokines. However, according to the available preclinical and clinical data, the most promising medications to be used for COVID-19 treatment are those that modulate platelets behaviour and blood coagulation factors, mainly through the P2Y12 receptor.

## 1. Introduction

In December 2019, a new single-stranded RNA virus from the Coronaviridae family emerged. It was named severe acute respiratory syndrome coronavirus 2 (SARS-CoV-2) and is the aetiological pathogen of coronavirus disease 2019 (COVID-19). Due to non-existent immunity at the time, the new virus quickly spread worldwide, progressing to a pandemic, as declared by the World Health Organization on 11 March 2020 [1]. Since 2021, vaccines and medications against the original form of SARS-CoV-2 have been approved, although their distribution has not been equal among countries. On the one hand, the United States has approved multiple vaccines and medications such as tocilizumab, remdesivir, and baricitinib. On the other hand, in low-income countries, vaccination protocols have been affected by the lack of the immunisation products, resulting in higher mortality rates.

SARS-CoV-2 enters a person through the upper respiratory tract. Infection of the host cell mainly occurs by the viral spike protein binding to the receptor-binding domain (RBD) of membrane-bound angiotensin-converting enzyme 2 (ace2). People infected with SARS-CoV-2 can develop asymptomatic, mild, moderate or severe forms of COVID-19. These last two stages manifest as an intense systemic inflammation characterised by the excessive release of pro-inflammatory molecules. These high levels cause tissue damage and dysregulate the coagulation cascade in multiple organs, leading to acute respiratory distress syndrome (ARDS), organ failure, and possibly the death of the infected patient [2,3].

One of the main players of the systemic inflammation manifested in COVID-19 is adenosine triphosphate (ATP). This pro-inflammatory component of the purinergic system is released during tissue damage and modulates other pathways, including cytokine release through the P2X7 receptor and the NLRP3 inflammasome activation [4]. Adenosine diphosphate (ADP), a product of ATP metabolism, can also exacerbate the disease by activating purine receptors on platelets, leading to thrombus formation [5]. 

These nucleotides and nucleosides are recognised by cells via the P2 and P1 purinergic receptors, which contain different subunits and have distinct affinities for these molecules, leading to different cellular effects [6]. The P2 receptors are divided into P2X1–P2X7, which respond exclusively to ATP, P2Y1–P2Y13 (which have affinity for both ATP and ADP), and pyrimidines (UTP and UDP). The P1 receptors—A1, A2A, A2B and A3—are responsive mainly to ADO, although they may also respond to INO [7]. On the other hand, concentrative nucleoside transporters (CNT1–CNT3) and equilibrative nucleotide transporters (ENT1–ENT4) help to regulate the transport of ADO between intracellular and extracellular spaces [8]. In the intracellular space, ADO can be converted into INO by cytoplasmic ADA or transformed into AMP by adenosine kinase (ADK), which results in the formation of ATP by adenylate kinase. This intracellular ATP can be released to extracellular space by transporter channels, catabolised to cyclic AMP (cAMP) by adenylate-cyclase and phosphodiesterase (PDE), or degraded to ADO by ATP/ADPase and CD73 [8] (Figure 1).

Many research groups, including ours, have hypothesised that purinergic signalling affects the course of COVID-19 infection by affecting blood flow, coagulation, and immunomodulation [9]. Some studies have indicated that the P1 and P2 receptors [10], pannexins [11], and CD39 and CD73 activity [12] influence the pathogenesis of COVID-19. As is the case with other infectious events, this disease causes damage to the host’s tissue due to the viral infection per se or because of exacerbated inflammation. With this injury, the levels of ATP in the extracellular space increase, consequently attracting more immune cells to the infection site and perpetuating the purinergic signalling cascade through the activation of its associated receptors and enzymes. Thus, the use of agonists and antagonists to these receptors and enzymes has been intensively evaluated for COVID-19 treatment.

Based on this background, in this review, we address the main preclinical and clinical scientific findings to date on pharmacological approaches targeting the purinergic signalling pathway as a rational alternative for COVID-19 treatment and its complications, such as blood coagulation, inflammation, vasodilation, and immunological processes, which are strongly affected during infection.

## 2. Methods 

As shown in Figure 2, we searched the PubMed and Scopus databases in addition to clinicaltrials.gov with the keywords ‘COVID-19’ AND ‘purinergic’. We used ‘purinergic’ as a keyword to collect all published papers about the specific subject. Variations such as ‘purinergic signalling’, ‘purinergic system’, or ‘purinergic pathway’ are used by researchers throughout the literature. The inclusion criteria were: (1) COVID-19 is the main subject of study, and (2) analysis of the purinergic signalling components as a target for therapy instead of prognosis. The exclusion criteria were: (1) reviews, comments or hypotheses; (2) duplicate papers; and (3) no experiments to test the hypotheses.

## 3. Results and Discussion

Our search yielded 53 papers from PubMed, 16 papers from Scopus (although 15 were duplicates of papers in PubMed, result in 1 paper), and 10 clinical trials from clinicaltrials.gov (Figure 2). Of these 54 papers, only eight experimentally targeted purinergic signalling components for COVID-19 treatment (Table 1). The remaining 45 papers are reviews, commentaries, perspectives, and suggestions, and did not involve experiments to prove the hypotheses; hence, we excluded them. Note that we searched clinicaltrials.gov by using the same keywords as we used to search the PubMed and Scopus databases. We found 10 clinical trials, but only 9 of these met the inclusion criteria. Table 2 provides the details of these trials.

We focus on highlighting the influence of the purinergic signaling cascade on the immune system and blood coagulation cascades, as both areas are a major concern due to their deregulation during SARS-CoV-2 infection. This imbalance in the immune system is, for the most part, linked to a cytokine storm, which is an exacerbated release of pro-inflammatory cytokines such as tumour necrosis factor alpha (TNF-α), interleukin 6 (IL-6), IL-1β, and others. This inflammatory explosion shapes immune cell behaviour and activates platelets and other proteins responsible for blood coagulation, resulting in dysregulation of events and immune thrombosis formation. Blood coagulation is directly linked to the activation and aggregation of platelets, resulting in the formation of blood clots, a low platelet count, higher D-dimer levels, elevated prothrombin activity time (TAP), and activated partial thromboplastin time (PTT) [19]. These clinical alterations can persist for 44–155 days after the onset of COVID-19 symptoms [20]. Hence, in 2021, the International Society on Thrombosis and Haemostasis (ISTH) recommended using low molecular–weight heparin to treat patients with moderate and severe COVID-19 [21]. Indeed, clinical trials have shown the benefits of treating patients with COVID-19 with anticoagulants. Higher doses of anticoagulants were associated with lower mortality in hospitalised patients with COVID-19 [22], and there was a dose-dependent delay in death due to COVID-19 [23]. These drugs are used to treat pathologies such as myocardial infarction, stroke, and arterial occlusive diseases [24,25]. 

Given the dysregulation of these events in patients with COVID-19, it is rational to target these purinergic receptors and enzymes. There has been preclinical and clinical research targeting the P2X3, P2X7, P2Y12, P2Y14, A2A, and A3A receptors, and the enzymes CD39 and CD73 (Table 1).

### 3.1. P2 Receptors in COVID-19

#### 3.1.1. P2Y Receptors

##### P2Y12

Based on our search, the P2Y12 receptor stands out as the most promising target for pharmacological modulation of COVID-19 to minimise one of the major culprits of this disease, namely, dysregulation of blood coagulation. Considering the positive results of anticoagulants, researchers began to evaluate the use of P2Y12 antagonists to manage patients with COVID-19. These antithrombotic agents, including clopidogrel, prasugrel, cangrelor, and ticagrelor, have antiplatelet and vasodilatory actions. 

The mechanism of action of those drugs involves P2Y12 receptor blockade on platelets, the main players in blood coagulation. Blood coagulation is initiated by cellular signalling of GpIb-IX-V, resulting in the secretion of agonists such as ADP. This nucleotide binds to P2Y1 and P2Y12 receptors, activating thromboxane A2 (TxA2) formation and cyclooxygenases (COXs), and thus triggering inflammatory processes and platelet activation and aggregation [26]. Therefore, P2Y12 receptor blockade attenuates the action of ADP on the formation of platelet aggregates, minimising the interaction with other platelet molecules, such as collagen and thrombin, among others [24].

Clopidogrel is a prodrug that is activated by metabolism to 2-oxo-clopidogrel and later the active thiol metabolite. After activation, clopidogrel irreversibly binds to the P2Y12 receptor on platelets and exerts a therapeutic effect lasting for 5–10 days, based on how long the platelet survives. Thus, this drug has the advantage of being long acting [25,27]. Prasugrel is also a prodrug; it is activated by specific cytochrome P450s (CYPs) [25]. Prasugrel begins working after 30 min, much faster than clopidogrel. This quicker bioavailability and platelet reactivation reduce the risk of extensive bleeding in patients [28]. Cangrelor and ticagrelor are ATP analogues with structural modifications to allow specific binding to the P2Y12 receptor. Cangrelor undergoes non-hepatic metabolism, while ticagrelor is metabolised in the liver by CYP3A4 [25,29]. These drugs have different administration routes: via a nasogastric tube for oral tablets for ticagrelor and intravenously for cangrelor. Cangrelor has a faster action and functional recovery of platelets, around 60–90 min, thus reducing the risk of major bleeding [25,30]. 

Table 2 lists clinical trials with P2Y12 antagonists that have been conducted in patients infected with SARS-CoV-2. Among them, clopidogrel stands out with five clinical trials investigating antithrombotic and antiplatelet actions (NCT04518735, NCT04368377, NCT04409834, NCT04505774 and NCT04333407) and evaluation of antiplatelet therapy in COVID-19 pneumonia (NCT02735707). 

NCT04409834 (COVID-PACT) evaluated the efficacy and safety of a prophylactic dose of anticoagulation and antiplatelet therapies. The study concluded that compared with placebo treatment, clopidogrel and four heparin variations may reduce all-cause mortality. However, these treatments have an uncertain influence on the necessity for additional respiratory support, COVID-19-related mortality, and quality of life [31]. Moreover, the trial showed that a full dose of an anticoagulant, except clopidogrel, reduced thrombotic complications in critically ill patients. There was an increase in bleeding in haemodynamically stable patients, but with no fatal outcomes [32]. 

NCT04368377 (PIC-19) analysed the prophylactic use of tirofiban, acetylsalicylic acid, clopidogrel, and fondaparinux. There was an improvement in peripheral oxygenation and a decrease in the need for mechanical respiratory support, with no adverse events reported [33]. In contrast, NCT04505774 (ACTIV-4A) showed that compared with heparin alone, clopidogrel together with heparin was not correlated with improvements in organ support–free days, up to 21 days, during hospitalisation [16].

NCT04333407 established the first endpoint of preventing cardiac complications of COVID-19, but it was terminated due to difficulty in recruiting eligible participants.

NCT04445623 is a phase III double-blind trial comparing the use of placebo with prasugrel in patients with COVID-19. The primary endpoint is to compare the efficiency index of pulmonary gas exchange. The partial pressure of oxygen arterial oxygen (PaO2) divides by the fraction of inspired oxygen (FiO2) known as PaO2/FiO2 ratio (PaO2/FiO2) or ROX index [34], and it was detected 7 days after treatment. The current status of this trial is unknown, and no results have been posted on clinicaltrials.gov. As mentioned for clopidogrel, NCT04505774 evaluated prasugrel together with heparin, but there was no improvement in organ support-free days [35].

A case report evaluating cangrelor and ticagrelor showed better results when compared with clopidogrel, mainly due to the reversibility of the antiplatelet effect [36]. Clopidogrel, cangrelor, and ticagrelor were analysed for COVID-19 treatment in NCT04518735, a retrospective study that enrolled 1707 participants. However, there are no publicly available results. NCT02735707 (the REMAP-CAP trial) enrolled more than 10,000 participants and compared several drugs for treatment of community-acquired pneumonia, including COVID-19. Among the multiple arms of the study, medications including clopidogrel, prasugrel, and ticagrelor were analysed. These medications did not improve the number of organ support-free days up to 21 days [37]. In vitro studies have shown that ticagrelor reduced the risk of secondary pulmonary infections and sepsis, possibly due to the influence on the immune system by decreasing IL-6 as well as neutrophil infiltration into the lungs [38].

Glucocorticoids have anti-inflammatory and immunosuppressive effects that can help reduce the severity of the cytokine storm, which is a key driver of severe COVID-19. Recently, omics-driven studies have shown the potential pleiotropic actions of synthetic glucocorticoids [38]. In addition, dexamethasone use in hospitalized COVID-19 patients without intensive respiratory support (IRS) did not show significant benefit and may even have potential harm, meaning that glucocorticoid therapy, such as dexamethasone, should be reserved for patients with severe or critical COVID-19 who require IRS [39]. Unlike glucocorticoids, P2Y12 receptor antagonists are more targeted and therefore they may have fewer off-target effects.

Taken together, there has been relatively little evidence that pharmacological modulation of the P2Y12 receptor reduces the severity of SARS-CoV-2 infection. Although the drugs are safe and have shown positive results in reducing prothrombotic complications and mortality, larger studies with more patients are needed to definitively determine their potential to treat COVID-19.

##### P2Y14

Although there are no records of clinical trials that have targeted P2Y14 to treat COVID-19, in vitro studies have shown that pharmacological modulation of the P2Y14 and A3 receptors as well as the deletion of the P2Y14R and P2X7R genes reduce cytokine levels and neutrophilia in a mouse model [37]. Both platelets and neutrophils are highly activated in COVID-19; neutrophil activation and neutrophil extracellular trap (NET) formation causes more thrombotic complications than platelet activation [40]. NET infiltration has a crucial role in infection due to the production of pro-inflammatory cytokines (IL-1β, IL-6, IL-8, TNF-α, monocyte chemoattractant protein-1 [MCP-1], and granulocyte-macrophage colony-stimulating factor [GM-CSF]), resulting in severe tissue damage, thrombus formation, vascular leakage, and potential necrosis. In SARS-CoV-2 infection, there is an increase in the levels of neutrophils and NET markers in patients with severe COVID-19 and those who die. These findings indicate a correlation between NETs and COVID-19 severity [41]. P2Y14 receptor antagonism may reduce neutrophil recruitment, minimising lung infiltration and the attenuating cytokine storm at the primary site of infection [42]. It is also worth mentioning that P2Y14 is an important proinflammatory receptor in other pathologies, such as ischaemic acute kidney injury [43], eosinophilic airway inflammation [44], and even glucose and oxygen deprivation in brain microvascular endothelial cells [45]. It also appears to have a great influence in diabetes, obesity, and even stem cell senescence [46,47,48]. Hence, therapies targeted to the P2Y14 receptor alongside pharmacological management of NET formation might represent a future approach to treat COVID-19.

#### 3.1.2. P2X Receptors

P2X receptors have been considered in the pathophysiology and pharmacological treatment of COVID-19. Unlike the P2Y12 receptor, there are no approved medications that directly act on P2X receptors. Our systematic review identified two preclinical studies that analysed P2X receptors. (Table 1).

##### P2X3 Receptor

Edwards et al. [17,18] showed the indirect effect of dexamethasone on the P2X3 receptor. Dexamethasone is a mineralocorticoid receptor inhibitor that suppresses cortisol secretion, thus preventing ATP release and subsequent P2X3 receptor activation. Activation of this purinergic receptor is linked to cough symptoms during COVID-19 and attenuation of this signalling cascade may result in reduction of this symptom [17,18]. While there is no clinical trial registered at clinicaltrials.gov testing this treatment alternative, a randomised clinical trial evaluating the P2X3 antagonist sivopixant in refractory chronic cough, not related to COVID-19, reported a symptom reduction [49]. Thus, this treatment alternative could also be investigated in COVID-19 with the aim of reducing symptoms and improving health-related quality of life among patients.

##### P2X7 Receptor

The role of the P2X7 receptor has been studied in the pathophysiology of many diseases, including COVID-19 [50]. Found in different cells and tissues, the P2X7 receptor has an important pro-inflammatory influence on immune cells and it facilitates the cytokine storm in COVID-19, which can persist for months even after recovery [50,51]. Recently, García-Villalba et al. [52] showed that the soluble P2X7 receptor concentration increases in blood plasma of patients with COVID-19, and this increase is positively correlated with disease severity and C-reactive protein levels, suggesting that this receptor could be a prognosis biomarker.

P2X7 receptor activation may lead to an influx of Ca^2+^ that stimulates the NLRP3 inflammasome, which is responsible for enhancing the release of pro-inflammatory cytokines, such as 1L-1β and IL-18, as well as caspase-1 activation that cleaves these cytokines into their mature and biologically active forms [51]. Other cytokines released through P2X7 receptor activation include lL-6, TNF-α, CCL2, IL-8, CCL3, and CXCL2 (Figure 3) [53,54]. Thus, researchers have suggested that P2X7 receptor activation can worsen COVID-19 [53], and the use of antagonists for this receptor could represent a strategy to attenuate the effects of its activation [50]. In this view, researchers have demonstrated that P2X7 receptor blockade or P2X7R gene deletion has direct effects on reducing the inflammatory state during the infectious process in an animal model [13].

Considering the importance of the P2X7 receptor in inflammation, we searched for medications that act on this receptor. We found that lidocaine shows partial modulation of the P2X7 receptor beyond its primary mechanism of action. There have been two clinical trials registered at clinitrials.gov that have evaluated the use of lidocaine for COVID-19 treatment. NCT04609865 (LidoCovid) is a phase III randomised clinical trial aiming to evaluate the effect of intravenous lidocaine 2% on gas exchange and inflammation in patients with ARDS related or not related to COVID-19 [55]. The other phase I randomised clinical trial, NCT04979923, will evaluate the efficacy of lidocaine nebulisation in cough suppression and amelioration of hypoxia in comparison with two other medications, salbutamol and beclomethasone. Although there are no results from these clinical trials, there is a case report from a 71-year-old man with COVID-19 and severe respiratory distress who received lidocaine intravenously. This treatment improved his inflammatory condition. As lidocaine was the only medication he received, this attenuation could be linked to modulation of the P2X7 receptor, reinforcing the potential of targeting this purinergic element directly or indirectly [56].

There are other medications with indirect effects on the P2X family of receptors. For example, ivermectin indirectly modulates the P2X4 receptor. However, the use of this medication as a prophylactic or therapeutic alternative for COVID-19 is controversial. Indeed, large randomised clinical trials and detailed reviews have already shown that this anti-helminthic medication has no benefit in the COVID-19 context [57,58,59,60,61].

### 3.2. P1 Receptors in COVID-19

In recent years, the P1 receptors have been extensively investigated as targets for drug development, mainly due to their immunosuppressive potential when activated by their main agonist ADO. These G protein–coupled receptors are divided into four isoforms—A1, A2A, A2B, and A3—each one with different affinities to ADO. The A1 and A2A receptors have high affinity to ADO (from 10 nm to 1 μm), while the A2B and A3 receptors have a low affinity to ADO (>10 μm). Once activated, it may inhibit (A1 and A3) or stimulate (A2A and A2B) adenylate cyclase (AC), leading to changes in cyclic AMP (cAMP) levels [8] (Figure 1).

#### The A1, A2A, A2B and A3 Receptors

Our search resulted in three preclinical and four clinical trials using therapies targeting P1 receptors. Among P1 receptors, the A2A receptor has been the most investigated as a therapeutic target for COVID-19. Indeed, this receptor is widely expressed in the pulmonary epithelium and modulates resident macrophages. It also prevents adherence and activation of neutrophils to the pulmonary epithelium [62,63]. The A2A receptor has a high affinity to ADO and stimulates AC, consequently increasing the intracellular cAMP levels and leading to complementary activation of anti-inflammatory mechanism. This mechanism is involved in the suppression of cytokine production and oxidising molecules, modulating neutrophils, macrophages, lymphocytes, and platelet aggregation [63]. Therefore, molecules acting on the A2A receptor may be a good alternative treatment for COVID-19. 

Tokano et al. [14] analysed the modulation of the A2A receptor with istradefylline in two preclinical studies (Table 1). Istradefylline is a selective A2A receptor antagonist indicated as an adjunct to levodopa and carbidopa for the treatment of Parkinson’s disease. In a recent study, the authors evaluated the effects of this drug on PBMCs. They showed suppression of IL-17A and IL-8 production and attenuation of consequent neutrophilic inflammation. As mentioned above, neutrophil activation and NET formation have a very strong influence on inflammation and severity in COVID-19. Therefore, targeting the A2A receptor with an antagonist could provide effective pharmacological modulation of COVID-19.

Another medication acting through the A2A receptor and evaluated as a treatment for COVID-19 is regadenoson. This molecule is a partial agonist of P1 receptors with affinity for the A2A receptor. It dilates coronary vessels and has been used as a diagnostic tool for cardiac pathologies [25]. This medication increases blood-flow in vessels and the myocardium and mimics the effects of ADO. NCT04606069 is currently recruiting participants to evaluate regadenoson in COVID-19; the results are expected in 2023.

In clinical practice, ADO is used to convert paroxysmal supraventricular tachycardia to sinus rhythm [25]. Because ADO acts on all four P1 receptors, it is not clear which receptors are responsible for this effect. For example, A1 receptor activation is related to monocyte phagocytosis, dendritic cell chemotaxis, and mucus promotion, increasing inflammation. In contrast, the A3 receptor has been touted to inhibit degranulation in eosinophils and neutrophils [64].

From this view, even with controversial results, clinical trials evaluating ADO in COVID-19 have been initiated and registered at clinicaltrials.gov (Table 2). NCT04588441 (the ARTIC trial) aims to measure the efficacy of aerosolised inhaled ADO against lung inflammation in patients with ARDS caused by COVID-19. This trial is not yet recruiting participants, and, therefore, no results have been released. On the other hand, a case-control study with the same treatment protocol was conducted in Italy. It showed that SARS-CoV-2-positive patients who received aerosolised ADO had an improved PaO2/FIO2 ratio, a reduced hospitalisation time and decreased SARS-CoV-2-positive days after diagnosis [65].

Moreover, as a possible role of the ADO receptors in COVID-19, cAMP regulation could be involved in the development of hyposmia and hypogeusia [66]. Lower cAMP levels are correlated with worsening of hyposmia and hypogeusia [67]. So, it seems that cAMP levels dictate the direction of the prognosis, because low levels worsen and high levels improve the ability to taste and smell [68]. 

Theophylline is a medication that can regulate cAMP concentration [69] and could increase the levels of this secondary messenger through ADO receptors and lead to olfactory neuroepithelium recovery in patients with COVID-19 [70]. To investigate this hypothesis, NCT04789499 evaluated the efficacy of theophylline in patients with symptoms of hyposmia and hypogeusia after SARS-CoV-2 infection. Even though 59% of the patients reported at least a slight improvement in their senses of smell and taste, there were no significant differences between the groups [71]. Another study had similar results in the recovery of smell and taste, in which patients treated with theophylline reported an improvement in smell compared with the placebo group [72]. Beyond the effects on hyposmia and hypogeusia, another pilot study showed that the administration of pentoxifylline and theophylline increased the efficiency of pulmonary gas exchange (PaO2/FiO2) and decrease C-reactive protein levels and mortality compared with the control group [73].

In addition to theophylline, other methylxanthines such as caffeine (1,3,7-trimethylxanthine) are natural compounds that act as P1 receptor antagonists. In silico analysis indicated that theophylline is a potential inhibitor of SARS-CoV-2 replication [74]. So, a molecule with a similar structure to caffeine could also have some beneficial effects on the pathology of COVID-19 by diminishing the cytokine storm and protecting the lungs from exacerbated inflammation [64]. Indeed, preclinical research has already demonstrated the potential of caffeine to inhibit the entry of SARS-CoV-2 into host cells by blocking the spike protein–ACE2 interaction [75]. Moreover, caffeine has been shown to improve oxygenation through relaxation in the pulmonary vascular muscles in chronic lung disease in premature cases, and improve lung function in asthma patients and patients with exercise-induced bronchoconstriction [64]. Clinical trials have been designed to prove the benefits of caffeine: there are 22 registered at clinicaltrials.gov with different outcome measures, from behavioural changes to molecular effects. NCT05594615 is a phase I trial that will evaluate drug–drug interaction between EDP-235 (SARS-CoV-2 antiviral), midazolam, caffeine, and rosuvastatin in healthy subjects. This trial will hopefully clarify whether caffeine can decrease the systemic inflammation manifested in COVID-19.

### 3.3. Purinergic Enzymes in COVID-19

Considering the importance of the ectoenzymes CD39 and CD73 in controlling extracellular nucleotides and nucleosides, several researchers have shown the correlation of their expression and function with COVID-19 development and severity [9,15,76,77]. Some metabolites derived from nucleotide metabolism are increased in blood samples of patients with COVID-19. Higher ADO levels are positively correlated with higher platelet counts [9]. Moreover, platelets from patients with COVID-19 show greater ATP, ADP, and AMP hydrolysis, and higher ADA activity, which could lead to higher INO blood concentrations [15]. Interestingly, the increase in INO is negatively correlated with lower white blood cell (WBC) and platelet counts, suggesting that high ADO levels could reduce WBC counts [9]. Therefore, the use of P2Y12 inhibitors could attenuate platelet activation and aggregation, reducing thrombus formation and the negative effects of the excessive nucleotide metabolism.

#### 3.3.1. CD39 and CD73

Although we did not include prognosis papers in this review, most of the papers we found from our search evaluated the prognostic value of CD39 and CD73 in COVID-19 (Table 1). For example, Da Silva et al. [15] analysed blood samples from patients with moderate and severe COVID-19, and confirmed the increased expression of both in total leucocytes based on the nucleotide hydrolysis activity. Indeed, Dorneles et al. [77] observed higher expression of CD73 (encoded by NT5E) and CD39 (encoded by ENTPD1) in T cells; the levels correlated with lower concentrations of ATP in blood plasma. However, another study observed lower expression of NT5E in the cytotoxic lymphocyte population of CD8 and natural killer T cells from patients with COVID-19, which secreted higher amounts of pro-inflammatory cytokines [12].

When searching clinicaltrials.gov, we did not find any clinical trials evaluating the modulation of CD39. However, two trials have been registered to evaluate monoclonal antibodies against CD73 in COVID-19, aiming to minimise ADO production [78]. NCT04516564, a phase I, randomised, double-blind, placebo-controlled trial is evaluating the safety, tolerability, pharmacokinetics, and immunogenicity of AK119 (monoclonal anti-CD73) in 29 healthy subjects. No results have been posted. The other phase I, non-randomised and open-label trial (NCT04464395) is testing mupadolimab (CPI-006), an anti-CD73 monoclonal antibody, in hospitalised patients with mild and moderate COVID-19. In this trial, the researchers have measured the efficacy, duration of COVID-19-related symptoms, hospitalisation time, rate of medical interventions during hospitalization, and changes in anti-SARS-CoV-2 immunoglobulin levels. The status of this study is listed as completed, but the results have not yet been released. Although there are no results yet, if anti-CD73 proves to be safe and tolerable, its use in patients with COVID-19 can be analysed in future clinical trials.

Due the multiple functions that CD73 has in inflammation, it is unclear what benefits CD73 inhibition would provide for COVID-19 treatment. Indeed, no publication has presented a clear explanation for this hypothesis. The only possible function we could assume so far is the modulation of cytotoxic lymphocytes described by Dorneles et al. [77], which could diminish the cytokine storm and inflammatory process of COVID-19. Even then, CD73 inhibition could cause the opposite effect because this enzyme is mainly anti-inflammatory (via the production of ADO). Therefore, more in vitro, in vivo, and clinical trials are needed to understand the real application of CD73 in COVID-19.

#### 3.3.2. PDEs and ADA

Researchers have also observed the pharmacological potential of modulating the intracellular enzymes of purinergic signalling. Zlamal et al. [79] showed that upregulation of intracellular cAMP levels could prevent the worsening of the prognosis and disease progression by attenuating immunoglobulin G–induced formation of procoagulant platelets. Targeting this pathway could minimise blood coagulation dysregulation. For example, dipyridamole is an inhibitor of nucleoside transporters, of PDE4A, PDE5A and PDE10A, and ADA, increasing the intracellular cAMP levels [80]. So, clinical trials using dipyridamole for COVID-19 treatment have been initiated. NCT04391179 and NCT04410328 have shown positive results based on the reduction of some inflammatory markers [81]. Therefore, targeting intracellular PDEs may be a good therapeutic choice for COVID-19 to avoid coagulation dysregulation because it prevents platelet aggregation and clot formation.

## 4. Conclusions

Purinergic signalling is directly linked to several physiological and pathological processes, such as inflammation, blood coagulation, and cellular signalling, which are affected in moderate and severe COVID-19. Although there is a limited amount of preclinical and clinical data evaluating the mechanism of action of medications targeting the purinergic system directly, there have been numerous indirect benefits for patients with COVID-19. These benefits indicate that there could be an advantage to using them mainly as a complementary treatment for COVID-19. We highlight the following:P2Y12 modulators such as cangrelor and ticagrelor could be the most promising medications due to their mechanism of action and reversible platelet blocking action, avoiding haemorrhagic events with excessive bleeding.P2Y14 is involved in neutrophil recruitment in COVID-19, and targeting this receptor may attenuate blood clot formation by minimising NET formation. However, no medication has been approved so far.Targeting the P2X3 receptor could relieve cough symptoms and perhaps improve quality of life.The P2X7 receptor is a promising target for inflammation reduction. Because this receptor is linked to NLRP3 inflammasome activation, blocking this element would reduce the release of pro-inflammatory cytokines such as IL-1 and IL-18.Targeting the ectoenzymes CD39 and CD73 does not seem to represent the best COVID-19 treatment strategy. If the CD39 enzyme is blocked and inactivated, ATP released by dying cells could concentrate in the extracellular space and chemoattract immune cells to the infection site, causing a loop of cytokine release resulting in tissue damage. On the other hand, blocking the enzyme CD73 would prevent the production of ADO, which could result in clinical improvements via activation of P1 receptors. However, using these enzymes in the same way as for prognostic biomarkers seems to be a good choice because measuring their expression and the levels of nucleotides can indicate the extent of tissue damage and the course of the disease.Targeting PDE and ADA intracellular enzymes could be an alternative treatment to avoid coagulation dysregulation and clot formation.Modulation of the A2A receptor with istradefylline and regadenoson represents a possible COVID-19 treatment because this receptor modulates neutrophils and the inflammatory process.The use of methylxanthines such as theophylline and caffeine could also be a good strategy in COVID-19 treatment due to their potential to help smell and taste recovery and to improve blood oxygen saturation.

Larger and well-designed studies are required to definitively assess whether medications targeting the purinergic system should serve as first-line or complementary treatment for COVID-19. Preclinical studies are necessary to confirm whether their mechanisms of action remain the same as the one originally approved, and clinical studies are crucial to evaluate their efficacy and safety in patients with COVID-19.

## Figures and Tables

**Figure 1 ijms-24-07865-f001:**
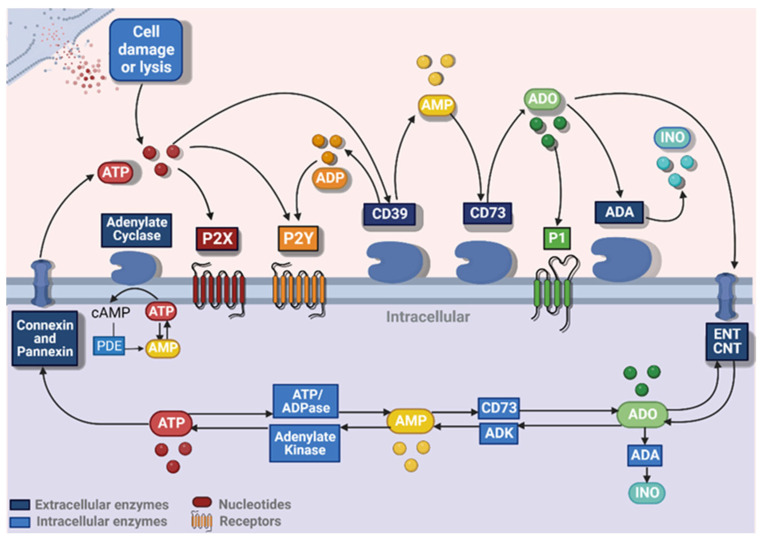
The purinergic signalling cascade. The release of ATP in the extracellular space leads to the activation of P2X and P2Y receptors and it is hydrolyzed by CD39 to ADP, which also activates P2Y receptors. The CD39 also hydrolyzes ADP to AMP, which is sequentially hydrolyzed to adenosine (ADO) by the CD73. ADO activates the P1 receptors and can return to the intracellular space by the ENT/CNT or be hydrolyzed by adenosine desaminase (ADA) into inosine (INO). Once the ADO is inside de cellular space, it can be converted to inosine (INO) by intracellular ADA or can be transformed into AMP by the Adenylate Kinase forming ATP by Adenylate cyclase (AC), which can be released into extracellular space by connexins and pannexins. The intracellular ATP can also be hydrolyzed to AMP by the intracellular ATP/ADPase and then again into ADO by CD73, being released again in the extracellular space by the CNT and ENT. Figure constructed using Biorender.

**Figure 2 ijms-24-07865-f002:**
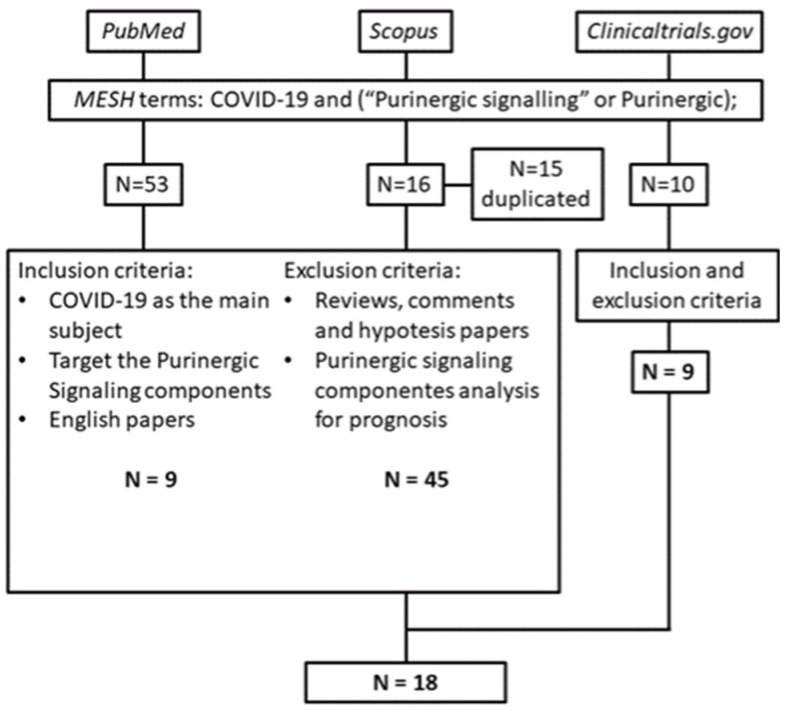
Visual abstract of methodology showing the database, keyword, inclusion and exclusion criteria, and final results from the systematic review.

**Figure 3 ijms-24-07865-f003:**
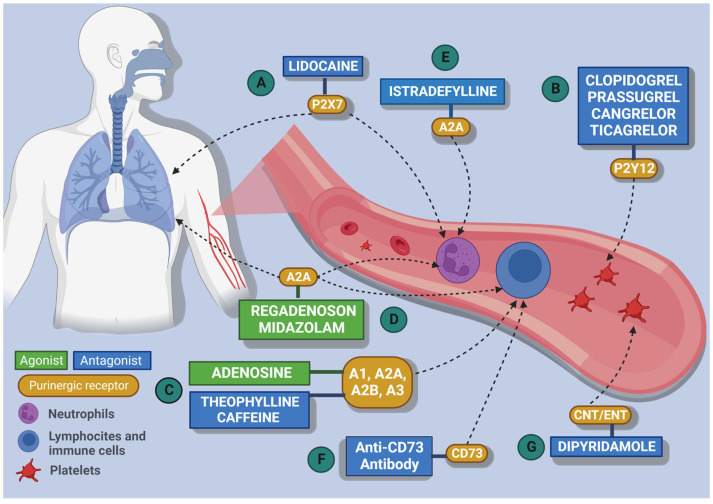
Schematic representation of the main targets of the drugs found in the systematic review. A. Receptor. The P2X7 antagonist, lidocaine act in the pulmonary environment, neutrophils, and immune cells. B. The P2Y antagonist (Clopidogrel, Prasugrel, Cangrelor, and Ticagrelor) act in the coagulation cascade by the P2Y12 receptors in platelets. C. Adenosine (ADO) is the main agonist of the A1, A2A, A2B, and A3 receptors, and, in contrast, theophylline and caffeine acts as an antagonist of these same receptors in immune cells. D. Regadenoson acts in the A2A receptor as an antagonist in immune cells and in the lung, where midazolam only acts as a potentiator of A2A receptor. E. Istradefylline acts as a antagonist in A2A receptors in neutrophils F. Anti-CD73 act as an enzymatic inhibitor in immune cells G. Dipyridamole acts as an antagonist of CNT and ENT transporters mostly in the platelets, culminating in the excess of ADO, which acts as an agonist in the P1 receptors (A1, A2A, A2B, A3) in the platelets, leading to vasodilation and antiplatelet aggregation effects. Figure constructed using Biorender.

**Table 1 ijms-24-07865-t001:** Articles results from the systematic review from the Scopus and PubMed database as described in the Methods section.

PMID	Title	Year of Publication	Purinergic Signalling Target	Drug	Purpose	Results
36268115	Effects of Purinergic Receptor Deletion or Pharmacologic Modulation on Pulmonary Inflammation in Mice [13]	2022	P2Y14R, P2X7R, 2Y14R and A3AR	P2Y14R, P2X7R genetic deletion and modulation with 2Y14R antagonists, A3AR agonists	Treatment	The extent of these responses was diminished by genetic deletion (P2Y14R, P2X7R) or pharmacologic modulation (P2Y14R antagonists, A3AR agonists) of purinergic receptors
35790489	Istradefylline, an adenosine A2a receptor antagonist, inhibits the CWHID4+ T-cell hypersecretion of IL-17A and IL-8 in humans [14]	2022	A2A	A2A antagonist (istradefylline)	Treatment	Attenuation of IL-8 and IL-17A release
35754396	Alterations in CD39/CD73 axis of T cells associated with COVID-19 severity [15]	2022	CD39 and CD73	Adenosine	Prognosis and Treatment	PBMC from severe COVID-19 patients treated with adenosine reduced the NF-κB activation in both CD3+ T cells and CD14+ monocytes. Lower levels of IL-1β and IL-17a were found in the culture supernatant of PBMC treated with adenosine, despite no changes in IL-10 and TNF-α production
35623041	Signalling via dopamine and adenosine receptors modulate viral peptide-specific and T-cell IL-8 response in COVID-19 [14]	2022	A2A	A2A antagonist (istradefylline)	Treatment	Attenuation of Il-8 release
35315874	Effect of Antiplatelet Therapy on Survival and Organ Support-Free Days in Critically Ill Patients with COVID-19: A Randomized Clinical Trial [13]	2022	P2Y12	Clopidogrel, prasugrel and ticagrelor;	Treatment	Among critically ill patients with COVID-19, treatment with an antiplatelet agent, compared with no antiplatelet agent, had a low likelihood of providing improvement in the number of organ support-free days within 21 days
35040887	Effect of P2Y12 Inhibitors on Survival Free of Organ Support Among Non-Critically Ill Hospitalized Patients with COVID-19: A Randomized Clinical Trial [16]	2022	P2Y12	Ticagrelor	Treatment	Among non-critically ill patients hospitalized for COVID-19, the use of a P2Y12 inhibitor in addition to a therapeutic dose of heparin, compared with a therapeutic dose of heparin only, did not result in increased odds of improvement in organ support-free days within 21 days during hospitalization.
34867791	Follow Your Nose: A Key Clue to Understanding and Treating COVID-19 [17]	2021	ATP	Dexamethasone and spironolactone	Treatment and Pathophysiology	Mineralocorticoid Receptor blockade can inhibit the release of ATP
33249452	New Horizons: Does Mineralocorticoid Receptor Activation by Cortisol Cause ATP Release and COVID-19 Complications? [18]	2021	Mineralocorticoid receptor, ATP and P2X3	Dexamethasone	Treatment	COVID-19 cough symptom is caused by the activation of purinergic receptors in the lungs following ATP release from virus-infected type II alveolar cells. This raises the question as to when treatment with dexamethasone and spironolactone should be started

**Table 2 ijms-24-07865-t002:** Results from the systematic review from the Clinical Trials.gov database as described in the Methods section.

**Purinergic Element**	**Pharmacological Mechanism**	**Medication**	**Original Therapeutic Use**	**Clinical Application on COVID-19**	**Clinical Trial**
P2Y12	Antagonist	Clopidogrel	Antiplatelet	Antiplatelet	NCT04333407 (N = 320) NCT02735707 (N = 10.000)
				Antithrombotic and antiplatelet	NCT04368377 (N = 5) NCT04505774 (N = 3.000) NCT04409834 (N = 390)
					NCT02735707 (N = 10.000)
	Antagonist	Prasugrel		Antiplatelet	NCT04445623 (N = 128)
				Antithrombotic and antiplatelet	NCT04505774 (N = 3.000)
	Inhibitor	Ticagrelor		Antiplatelet	NCT02735707 (N = 10.000)
A2A	Agonist	Regadenoson	Vasodilator used in radionuclide myocardial perfusion imaging	Anti-inflammatory	NCT04606069 (N = 40)
A1, A2A, A2B	Antagonist	Theophylline	Treat airflow obstruction associated with chronic lung diseases		NCT04789499 (N = 62)
PDE3A *, PDE4A *, PDE5A *					
A1, A2A, A2B, A3	Agonist	Adenosine	Myocardial perfusion scintigraphy and converts sinus rhythm of paroxysmal supraventricular tachycardia		NCT04588441 (N = 30)
A1, A2A, A2B, A3	Antagonist	Caffeine	Stimulant, and prevents and treats pulmonary complications of premature birth		NCT05594615 (N = 24)
PDE enzymes *					
A2A	Potentiator	Midazolam	Benzodiazepine with rapid onset that is used in seizures, anesthesia, and anxiety disorders		

* Biological targets that also are trigger by the respective drugs.

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
