# Peer review of "A Systematic Review of the Role of Purinergic Signalling Pathway in the Treatment of COVID-19"

_ijms, 2023, doi:10.3390/ijms24097865_

Round 1

Reviewer 1 Report

Hi 

Thank you for touching into important and unexplored issue in COVID-19. 

Despite the interesting topic, the manuscript fail in multiple areas

- The manuscript is just a compilation of manuscripts without a clear point of view from the authors. A scientific communication needs a scientific comment or suggestions. THIS IS A MAJOR ISSUE

- Abstract, needs a full reorganization. Please do not add details of methods and problems 

- Multiple repetitions and miss spellings all over 

- Tables are de-organized and sometime meaningless, because the authors include manuscripts not consider in the text. Why?

-References are not fully inclusive and needs a better description

Author Response

We are very grateful to the reviewers for their analysis of our work and for their constructive comments, which helped us improve the quality of our manuscript.

Reviewer #1 

Thank you for touching into important and unexplored issue in COVID-19. 

Specific comments:

R#1: Despite the interesting topic, the manuscript fails in multiple areas. The manuscript is just a compilation of manuscripts without a clear point of view from the authors. A scientific communication needs a scientific comment or suggestions. THIS IS A MAJOR ISSUE.

Answer: Thank you very much for your careful analysis of our work. We agree with the reviewer that there was a limitation in the discussion about the studies described in this review. To adapt the text to your suggestions, we completely reformulated each topic and inserted more information about the studies that already have published results, detailing the main scientific findings and their impact on the treatment of COVID-19, as well as our point of view. However, some clinical trials do not yet have published scientific results, which limited us in furthering the discussion in some sections.

R#1: Abstract, needs a full reorganization. Please do not add details of methods and problems .

Answer: We made a substantial reorganization of the abstract. We removed the information about the methodology used, updated some information, and added a final sentence about what would be the best purinergic pathways to be evaluated in the treatment of COVID-19.

“The coronavirus disease 2019 (COVID-19) pandemic, caused by severe acute respiratory syndrome coronavirus 2 (SARS-CoV-2), has become a global health concern. Three years since its origin, despite the approval of vaccines and specific treatments against this new coronavirus, there are still high rates of infection, hospitalisation and mortality in some countries. COVID-19 is characterised by a high inflammatory state and coagulation disturbances that may be linked to purinergic signalling elements such as adenosine triphosphate (ATP), adenosine diphosphate (ADP) and purinergic receptors. Therefore, drugs targeting this purinergic pathway, currently used for other pathologies, are being evaluated in preclinical and clinical trials for COVID-19. In this review, we focus on the potential of these drugs that control the release, degradation and reuptake of these extracellular nucleotides and nucleosides to treat COVID-19. These nucleotides/nucleosides play important roles in cellular processes, such as immunomodulation, blood clot formation and vasodilation, which are affected during SARS-CoV-2 infection. Thus, pharmacological strategies modulating this signalling pathway have shed light on possible strategies to treat this disease. According to the available preclinical and clinical data, the most promising medications to be used for COVID-19 treatment are those that modulate platelets behaviour and blood coagulation factors, mainly through the P2Y12 receptor. Drugs targeting the P2X7 and P1 receptors could also have therapeutic efficacy due to their capacity to modulate the cytokine storm and the immune response.”

R#1: Multiple repetitions and miss spellings all over.

Answer:  Thank you for your observation. After extensive grammar and English proofreading by the English editing service, duplicate information has been removed and reorganized within the text.

For example, the information in the sentence below was repeated in the P2Y12 receptor section and has been removed there.

 “Indeed, clinical trials have shown the benefits of treating patients with COVID-19 with anticoagulants. Higher doses of anticoagulants were associated with lower mortality in hospitalised patients with COVID-19 [18], and there was a dose-dependent delay in death due to COVID-19 [19]. These drugs are used to treat pathologies such as myocardial infarction, stroke and arterial occlusive diseases [20,21].”

Similarly, information about the mechanism of action of drugs on the P2Y12 receptor was repeated in a previous paragraph. The information has been retained only on page 8 (lines 180-183) and page 9 (lines 194-197).

“The mechanism of action of those drugs involves P2Y12 receptor blockade on platelets, the main players in blood coagulation. Blood coagulation is initiated by cellular signalling of GpIb-IX-V, resulting in the secretion of agonists like ADP. This nucleotide binds to P2Y1 and P2Y12 receptors, activating thromboxane A2 (TxA2) formation and cyclooxygenases (COXs) and thus triggering inflammatory processes and platelet activation and aggregation [22]. Therefore, P2Y12 receptor blockade attenuates the action of ADP on the formation of platelet aggregates, minimising the interaction with other platelet molecules, such as collagen and thrombin, among others [20].”

R#1: Tables are de-organized and sometime meaningless because the authors include manuscripts do not consider in the text. Why? 

Answer: Thank you for your careful analysis. In fact, the tables had some wrong information. We have evaluated all the inserted articles and reorganized the tables.

In table 1 we left only the nine data articles selected by the inclusion and exclusion criteria used for the systematic review. To facilitate the readers' understanding, we included a column with a brief explanation of the observed response.

In table 2 we put the clinical trials with drugs that act through purinergic signaling that are ongoing or completed. These clinical trials were also selected using the inclusion and exclusion criteria of the systematic review.

In the previous version of the tables there had been clinical and preclinical studies that did not meet the inclusion and exclusion criteria of the systematic review, especially in the criterion of the components of purinergic signaling being targeted for therapy rather than prognostic. In this way, we also chose to exclude table S1 as it presented articles that did not meet the inclusion and exclusion criteria and could cause confusion in the understanding of the systematic review.

However, to improve the discussion in some sections we have retained some prognostic articles, which are not included in the tables. These additional articles were important to support the discussion on some clinical trials and are included in the bibliography.

R#1: References are not fully inclusive and needs a better description.

Answer: We reorganized all the references and added them to the text with the help of Mendeley software, which was adjusted to insert the information according to the journal's rules.

Reviewer 2 Report

I read your manuscript with interest. My recommendation is to revise the article to enhance its readability. 

1. Consider re-writing the title as “A systematic review of the role of purinergic signalling pathway in the treatment of COVID-19.”

2. The introduction is winding and not presented logically. Present paragraphs in a way that one naturally leads to and connects with the next.

3. Much of the historical background about COVID-19, vaccines etc, could be easily summarized in one paragraph

4. Substantial English language editing is needed

5. There is no table showing the reviewed papers

6. Please confirm the PRISMA reporting style was followed.

7. In the discussion, the opening paragraph should have answered the question ‘what is the purinergic system’ before going ahead with the detailed explanations.

8. Under P2 Receptors in COVID-19, the presentation was mostly narrated; data from studies should have been systematically analyzed and discussed.

9. For clarity, consider using subheadings in the discussion part.

10. The tone is a bit too much like a traditional review. A systematic review should synthesize data and come up with clear conclusions. A concise summary to tie all the points together is lacking.

11. The conclusion should be written to present the main points from the reviewed papers

Author Response

RESPONSE TO REVIEWERS

We are very grateful to the reviewers for their analysis of our work and for their constructive comments, which helped us improve the quality of our manuscript.

Reviewer #2 

I read your manuscript with interest. My recommendation is to revise the article to enhance its readability.  

Specific comments:

R#2: Consider re-writing the title as “A systematic review of the role of purinergic signalling pathway in the treatment of COVID-19.” 

Answer: Thank you for your suggestion and we agree that this title better represents the goal of the systematic review. Therefore, we have changed the title to the new one suggested by the reviewer.

R#2: The introduction is winding and not presented logically. Present paragraphs in a way that one naturally leads to and connects with the next. Much of the historical background about COVID-19, vaccines etc, could be easily summarized in one paragraph.

Answer:  Thank you very much for the careful reading of our manuscript. To make the introduction easier to read, we have rewritten some sentences and deleted some information that we have detected as unnecessary for understanding the topic addressed. In addition, the manuscript was sent to an English editing service that made many adjustments to make the sentences more connected and integrative.

R#2: Substantial English language editing is needed

Answer: The manuscript was sent to an English editing service for extensive proofreading and correction of grammatical errors. The certificate was attached to the journal's system.

R#2: There is no table showing the reviewed papers.

Answer: As mentioned for R#1, we corrected tables 1 and 2, and deleted table S1 as it did not add pertinent information.

R#2: Please confirm the PRISMA reporting style was followed.

Answer: We reorganized the article and results following the most recent instructions of the PRISMA guideline published by Page et al. 2021. (Page, M.J., McKenzie, J.E., Bossuyt, P.M. et al. The PRISMA 2020 statement: an updated guideline for reporting systematic reviews. Syst Rev 10, 89 (2021).

R#2: In the discussion, the opening paragraph should have answered the question ‘what is the purinergic system’ before going ahead with the detailed explanations. 

Answer: Thanks for your suggestion. However, throughout the course of correcting the review to make it easier to read, as well as adding pertinent information that was missing, we noted that it would be better to keep the explanation of what purinergic signaling is in the introduction, allowing important concepts of this pathway to be understood in the start of reading.

This information is located on page 3

“Since its discovery in the early 19th century, the role of ATP and its degradation products has been studied due to its participation in several cell functions including neurotransmission; vascular remodelling; blood flow regulation; control of cell growth, proliferation and death; and immunomodulation [6]. Purinergic signalling is an important pharmacological target for many pathologies [7]. This signalling pathway involves a complex cascade of release, production, degradation and reuptake of nucleotides and nucleosides in the intracellular and extracellular spaces. As demonstrated in Figure 1, the release of ATP into the extracellular space may occur due to cell damage and lysis or by hemichannels (pannexins and connexins). Once in the extracellular space, this nucleotide is degraded by several enzymes, such as ectonucleoside triphosphate diphosphohydrolase (E-NTPDases) and ectonucleotide pyrophosphatase/phosphodiesterases (E-NPPs), to ADP and adenosine monophosphate (AMP). Ecto-5'-nucleotidase (CD73) hydrolyses AMP to adenosine (ADO), which may be hydrolysed by alkaline phosphatase and adenosine deaminase (ADA) to inosine (INO). Moreover, ADO can also be generated by a non-canonical pathway: AMP is produced from NAD+ via the enzymes CD38 and ENPP1, and ADO is generated by CD73 [8].

These nucleotides and nucleosides are recognised by cells via the P2 and P1 purinergic receptors, which contain different subunits and have distinct affinities for these molecules, leading to different cellular effects [9]. The P2 receptors are divided into P2X1–P2X7, which respond exclusively to ATP, and P2Y1–P2Y13), which have affinity for both ATP and ADP, as well as to pyrimidines (UTP and UDP). The P1 receptors – A1, A2A, A2B and A3 –are responsive mainly to ADO, although they may also respond to INO [10]. On the other hand, concentrative nucleoside transporters (CNT1–CNT3) and equilibrative nucleotide transporters (ENT1–ENT4) help to regulate the transport of ADO between intracellular and extracellular spaces [8]. In the intracellular space, ADO can be converted into INO by cytoplasmic ADA or transformed into AMP by adenosine kinase (ADK), which results in the formation of ATP by adenylate kinase. This intracellular ATP can be released to extracellular space by transporters channels, catabolised to cyclic AMP (cAMP) by adenylate-cyclase and phosphodiesterase (PDE), or degraded to ADO by ATP/ADPase and CD73 [8] (Figure 1).”

R#2: Under P2 Receptors in COVID-19, the presentation was mostly narrated; data from studies should have been systematically analyzed and discussed. 

Answer:We appreciate your suggestion and agree that we were missing a more detailed discussion about the results of the mentioned articles. We analyze in detail the results of the studies already published and add throughout the sections. In addition, in some sections we have added our point of view on the observed results and perspectives in the treatment of COVID-19.

However, in some sections due to the reduced number of articles found by the systematic review and there being only clinical trials in the recruitment phase or in progress, it was not possible to have an intense discussion on the subject.

R#2: For clarity, consider using subheadings in the discussion part. 

Answer: Thanks for the suggestion. We have added subheadings throughout the manuscript to make it easier to read.

R#2: The tone is a bit too much like a traditional review. A systematic review should synthesize data and come up with clear conclusions. A concise summary to tie all the points together is lacking. 

Answer: Thank you very much for your careful analysis of our work. We agreed with the reviewer that the manuscript was in a more traditional style than necessary for a systematic review. From the rewording of the text for correction of English and other points mentioned by the reviewers, we also added paragraphs with short summaries about the clinical and preclinical studies added.

Below is an example of short descriptions we made about the inserted studies:

Page 9, lines 217-224: “NCT04409834 (COVID-PACT) evaluated the efficacy and safety of a prophylactic dose of anticoagulation and antiplatelet therapies. The study concluded that compared with placebo treatment, clopidogrel and four heparin variations may reduce all-cause mortality. However, these treatments have an uncertain influence on the necessity for additional respiratory support, COVID-19-related mortality and quality of life [27]. Moreover, the trial showed that a full dose of an anticoagulant, except clopidogrel, reduced thrombotic complications in critically ill patients. There was an increase in bleeding in haemodynamically stable patients, but with no fatal outcomes [28].”

R#2: The conclusion should be written to present the main points from the reviewed papers. 

Answer: We appreciate your suggestion. From this we change our conclusion and add a sentence about the main results observed in each topic.

“Conclusion: Purinergic signalling modulates several physiological and pathological processes, such as inflammation, blood coagulation and cellular signalling, which are affected in moderate and severe COVID-19. Although there is a limited amount of preclinical and clinical data evaluating the mechanism of action of medications targeting the purinergic system directly, there have been numerous indirect benefits for patients with COVID-19. These benefits indicate that there could be an advantage of using them mainly as complementary treatment for COVID-19. We highlight the following:

  • P2Y12 modulators such as cangrelor and ticagrelor could be the most promising medications due to their mechanism of action and reversible platelet blocking action, avoiding haemorrhagic events with excessive bleeding.
  • P2Y14 is involved in neutrophil recruitment in COVID-19 and targeting this receptor may attenuate blood clot formation by minimising NET formation. However, no medication has been approved so far.
  • Targeting the P2X3 receptor could relieve cough symptoms and perhaps improve quality of life.
  • The P2X7 receptor is a promising target for inflammation reduction. Because this receptor is linked to NLRP3 inflammasome activation, blocking this element would reduce the release of pro-inflammatory cytokines such as IL-1 and IL-18.
  • Targeting the ectoenzymes CD39 and CD73 does not seem to represent the best COVID-19 treatment strategy. If the CD39 enzyme is blocked and inactivated, ATP released by dying cells could concentrate in the extracellular space and chemoattract immune cells to the infection site, causing a loop of cytokine release resulting in tissue damage. On the other hand, blocking the enzyme CD73 would prevent the production of ADO, which could result in clinical improvements via P1 receptors. However, using these enzymes as for prognostic biomarkers seems to be a good choice because measuring their expression and the levels of nucleotides can indicate the extent of tissue damage and the course of the disease.
  • Targeting intracellular enzymes could be an alternative treatment to avoid coagulation dysregulation and clot formation.
  • Modulation of the A2A receptor with istradefylline and regadenoson represents a possible COVID-19 treatment because this receptor modulates neutrophils and the inflammatory process.
  • The use of methylxanthines such as theophylline and caffeine could also be a good strategy in COVID-19 treatment due to their potential to help smell and taste recovery and to improve blood oxygen saturation.

Larger and well-designed studies are required to definitively assess whether medications targeting the purinergic system should serve as first-line or complementary treatment for COVID-19. Preclinical studies are necessary to confirm whether their mechanisms of action remain the same as the one originally approved, and clinical studies are crucial to evaluate their efficacy and safety in patients with COVID-19.”

Reviewer 3 Report

Peer Review

In the manuscript “Targeting the purinergic signalling pathway as treatment for COVID-19: a systematic review” the authors discuss drugs acting through purinergic signalling by a systematic review. This is an important theme given the roles of immune response, blood clot formation, and vasodilation in COVID-19 infection, all of which can be modulated through purinergic signalling pathways. The manuscript as written contains interesting work and materials, but would benefit from the conclusions being stated more clearly. The format of the manuscript is also inadequate, and would be easier for the reader to absorb if it had a supplementary materials section.

Abstract

The paper as written does not bring its own conclusions to the fore. Having done a systematic review, the authors should be well placed to comment directly on which treatments targeting purinergic signalling show the most promise, and which are less promising. Your conclusions are helpful: specifically “Regarding clinical trials, those testing the P2Y12 antagonists (Clopidogrel, prasugrel, cangrelor, ticagrelor) are the most promising due to their function and specificity of attenuating platelets activation and coagulation” and this should be in the Abstract.

Introduction

Line 66: “allowing the allowing”, doesn’t make sense

Line 69: “intense systemic inflammation and coagulation disturbs in multiple organs” would be better written as “intense systemic inflammation and dysregulation of the coagulation cascade in multiple organs”

Line 83: is this correct? FDA has approved Remdesvir, Baricitinib AND Tocilizumab. Please check and amend if necessary.

https://www.fda.gov/drugs/emergency-preparedness-drugs/coronavirus-covid-19-drugs

Methodology

Are your search terms correctly described? What does searching for “Purinergic signaling” find that searching for “Purinergic” does not. Surely searching for the latter will find everything from the former?

Consider providing a summary for the reader of the number of participants in the trials / reports that you cover.

Table 1 is not appropriate. The full detail can be better presented in Supplementary Material, not the main body of the text. This table spans 17 pages. Consider putting it entirely in supplemental and preparing a single-page table that includes only the 9 studies. Also, Table 1 is the Results of your method, it is not the method. The summary version of Table 1 should therefore be placed in Results.

Line 168, “… 09 attended the inclusion criteria”. Please remove the unnecessary zero and replace attended with met, i.e. “… 9 met the inclusion criteria”

Results & Discussion

Table 1 should be included here in summary form. Table 2 is again inappropriately placed. Consider including the full table in Supplementary Material and a summary version in the main text.

Why is bold text used for some trials (line 261) but not others (line 221)?

Line 223, "The NCT04518735 is a retrospective, observational, one-center ..." Remove The at the start of the sentence, it is unnecessary.

Line 281, “receptor., Tand therefore” is a typographical error, please correct.

Lines 289-295, the authors could helpfully reference the pleiotropic effects of corticosteroids in the case of COVID-19 – whilst this class of drugs can attenuate the inflammatory response, corticoids are an extremely blunt instrument and are not indicated for use except in cases requiring supplemental oxygen (see NICE guidelines and for example PMID: 36292938 and PMID: 34824060). This reinforces your later point that P2Y12 antagonists are the most promising (and are clearly more targeted than corticosteroids, whose effects are essentially indirect rather than direct, and where the mechanisms are by now well-described).

PMID: 35040887 reported the results from the clinical comparison between Ticagrelor and Clopidogrel, showing broad equivalence in outcomes. This reference could usefully be added.

Conclusions

Line 602, suggest rewording the second sentence in the second paragraph to: “Given that these are the most affected areas during SARS-CoV-2 infection, the repositioning of these medications to treat moderate and severely ill patients has the potential to improve outcomes.” As currently written, the text is too vague.

Line 612, suggest rewording to “.. specificity of attenuating platelets activation and the dysregulation of coagulation”

Author Response

RESPONSE TO REVIEWERS

We are very grateful to the reviewers for their analysis of our work and for their constructive comments, which helped us improve the quality of our manuscript.

Reviewer #3 

Reviewer #3 

In the manuscript “Targeting the purinergic signalling pathway as treatment for COVID-19: a systematic review” the authors discuss drugs acting through purinergic signalling by a systematic review. This is an important theme given the roles of immune response, blood clot formation, and vasodilation in COVID-19 infection, all of which can be modulated through purinergic signalling pathways. The manuscript as written contains interesting work and materials, but would benefit from the conclusions being stated more clearly. The format of the manuscript is also inadequate, and would be easier for the reader to absorb if it had a supplementary materials section. 

Specific comments:

R#3: Abstract: The paper as written does not bring its own conclusions to the fore. Having done a systematic review, the authors should be well placed to comment directly on which treatments targeting purinergic signalling show the most promise, and which are less promising. Your conclusions are helpful: specifically “Regarding clinical trials, those testing the P2Y12 antagonists (Clopidogrel, prasugrel, cangrelor, ticagrelor) are the most promising due to their function and specificity of attenuating platelets activation and coagulation” and this should be in the Abstract. 

Answer: We are grateful for the thorough review of our manuscript. These limitations were pointed out by other reviewers and as described earlier to correct these shortcomings we have rewritten several sections to make them easier to read, better describe the results of each study add and address our view on the conclusions of the subject.

We also added a conclusion in the abstract on the most promising drugs for the treatment of COVID-19:

Page 1, lines 15-20: “According to the available preclinical and clinical data, the most promising medications to be used for COVID-19 treatment are those that modulate platelets behaviour and blood coagulation factors, mainly through the P2Y12 receptor. Drugs targeting the P2X7 and P1 receptors could also have therapeutic efficacy due to their capacity to modulate the cytokine storm and the immune response.”

R#3: Introduction 

R#3: Line 66: “allowing the allowing”, doesn’t make sense 

R#3: Line 69: “intense systemic inflammation and coagulation disturbs in multiple organs” would be better written as “intense systemic inflammation and dysregulation of the coagulation cascade in multiple organs” 

R#3: Line 83: is this correct? FDA has approved Remdesvir, Baricitinib AND Tocilizumab. Please check and amend if necessary. https://www.fda.gov/drugs/emergency-preparedness-drugs/coronavirus-covid-19-drugs 

Answer: We agreed that the manuscript needed an intensive English revision to avoid these poorly understood sentences. For this reason, we submitted the manuscript to an English editing service, which corrected and improved the writing of the manuscript. We believe that these writing errors have all been resolved.

Regarding the information on drugs approved for the treatment of COVID-19, all of these drugs mentioned are on the list of drugs approved for the treatment of children and/or adults of COVID-19: https://www.fda.gov/drugs/emergency-preparedness-drugs/coronavirus-covid-19-drugs

Actemra (Tocilizumab) is approved for the treatment of COVID-19 in hospitalized adults who are receiving systemic corticosteroids and require supplemental oxygen, non-invasive or invasive mechanical ventilation, or extracorporeal membrane oxygenation (ECMO).

Veklury (Remdesivir) is approved for the treatment of COVID-19 in adults and pediatric patients (28 days of age and older and weighing at least 3 kilograms) with positive results of direct SARS-CoV-2 viral testing, who are: hospitalized, or not hospitalized and have mild-to-moderate COVID-19 and are at high risk for progression to severe COVID-19, including hospitalization or death.

Olumiant (baricitinib) is approved for the treatment of COVID-19 in hospitalized adults requiring supplemental oxygen, non-invasive or invasive mechanical ventilation, or extracorporeal membrane oxygenation (ECMO).

R#3: Methodology 

R#3: Are your search terms correctly described? What does searching for “Purinergic signaling” find that searching for “Purinergic” does not. Surely searching for the latter will find everything from the former? 

Answer: Thanks. We agree that the description of the methodology used was a bit confusing. In this way, we rewrote and detailed how the search for the keywords was carried out, emphasizing the reason why only the term purinergic was sufficient to carry out the bibliographic search.

Page 4, lines 110-114: “As shown in Figure 2, we searched the PubMed and Scopus databases in addition to clinicaltrials.gov with the keywords ‘COVID-19’ AND ‘purinergic’. We used ‘purinergic’ as a keyword to collect all published papers about the specific subject. Variations such as ‘purinergic signalling’, ‘purinergic system’ or ‘purinergic pathway’ are used by researchers throughout the literature.”

R#3: Consider providing a summary for the reader of the number of participants in the trials / reports that you cover.

Answer: We appreciate the suggestion and agree that the addition of this information will make the manuscript easier to read. This information was added in each clinical trial included in the systematic review.

R#3: Line 168, “… 09 attended the inclusion criteria”. Please remove the unnecessary zero and replace attended with met, i.e. “… 9 met the inclusion criteria” 

Answer: Thank you for your observation. We have corrected the sentence.

R#3: Table 1 is not appropriate. The full detail can be better presented in Supplementary Material, not the main body of the text. This table spans 17 pages. Consider putting it entirely in supplemental and preparing a single-page table that includes only the 9 studies. Also, Table 1 is the Results of your method, it is not the method. The summary version of Table 1 should therefore be placed in Results. 

Answer:Thank you for your suggestions. The correction of the tables was also suggested by the other reviewers. As described earlier the tables had some wrong information. We have evaluated all the inserted articles and reorganized the tables.

            As suggested by you, we changed table 1 to contain only the nine articles selected for the systematic review and table 2 to contain only the clinical trials, with their respective information. We chose to exclude table S1 as it presented articles that did not meet the inclusion and exclusion criteria and could cause confusion in the understanding of the systematic review.

As suggested, we have also added a summary of the results obtained from the bibliographic search at the beginning of the results section:

Page 4, lines 120-128: Our search yielded in 53 papers from PubMed, 16 papers from Scopus (although 15 were duplicates of papers in PubMed, resulting in 1 paper) and 10 clinical trials from clinicaltrials.gov (Figure 2). Of these 54 papers, shown in Table S1, only nine experimentally targeted purinergic signalling components for COVID-19 treatment (Table 1). The remaining 45 papers are reviews, commentaries, perspectives and suggestions and did not involve experiments to prove the hypotheses; hence, we excluded them. Note that we searched clinicaltrials.gov by using the same keywords as we used to search the PubMed and Scopus databases. We found 10 clinical trials, but only 9 of these met the inclusion criteria. Table 2 provides the details of these trials.

R#3: Results & Discussion 

R#3:Table 1 should be included here in summary form. Table 2 is again inappropriately placed. Consider including the full table in Supplementary Material and a summary version in the main text. 

Answer: As described in the previous answer, adjustments were made to all tables to resolve these issues.

R#3:Why is bold text used for some trials (line 261) but not others (line 221)? 

Answer: Thank you for these observations. We removed all words in bold and left only the indication of figures and tables in bold.

R#3: Line 223, "The NCT04518735 is a retrospective, observational, one-center ..." Remove The at the start of the sentence, it is unnecessary. 

Line 281, “receptor., Tand therefore” is a typographical error, please correct. 

Answer:Thank you for careful analysis of the manuscript. All grammar errors were corrected after the extensive review of English.

R#3: Lines 289-295, the authors could helpfully reference the pleiotropic effects of corticosteroids in the case of COVID-19 – whilst this class of drugs can attenuate the inflammatory response, corticoids are an extremely blunt instrument and are not indicated for use except in cases requiring supplemental oxygen (see NICE guidelines and for example PMID: 36292938 and PMID: 34824060). This reinforces your later point that P2Y12 antagonists are the most promising (and are clearly more targeted than corticosteroids, whose effects are essentially indirect rather than direct, and where the mechanisms are by now well-described). 

PMID: 35040887 reported the results from the clinical comparison between Ticagrelor and Clopidogrel, showing broad equivalence in outcomes. This reference could usefully be added. 

Answer: We appreciate your suggestion and included the following paragraph in the end of P2Y12 topic:

“Glucocorticoids have anti-inflammatory and immunosuppressive effects that can help reduce the severity of the cytokine storm, which is a key driver of severe COVID-19. Recently, omics-driven studies have shown the potential pleiotropic actions of synthetic glucocorticoids (Int. J. Mol. Sci. 2022, 23, 12079. https://doi.org/10.3390/ijms232012079). In addition, dexamethasone use in hospitalized COVID-19 patients without intensive res-piratory support (IRS) did not show significant benefit and may even have potential harm, meaning that glucocorticoid therapy, such as dexamethasone, should be reserved for pa-tients with severe or critical COVID-19 who re-quire IRS (Eur Respir J 2022; 60: 2102532  https://doi.org/10.1183/13993003.02532-2021 ). Unlike glucocorticoids, P2Y12 receptor antagonists are more targeted and therefore, they may have fewer off-target effects.”

The reference PMID: 35040887 was included in Table 1.

R#3: Conclusions 

Line 602, suggest rewording the second sentence in the second paragraph to: “Given that these are the most affected areas during SARS-CoV-2 infection, the repositioning of these medications to treat moderate and severely ill patients has the potential to improve outcomes.” As currently written, the text is too vague. 

Line 612, suggest rewording to “.. specificity of attenuating platelets activation and the dysregulation of coagulation” 

Answer:Thank you.The correction of the conclusion section was suggested by other reviewers, so we completely rewrote this section and these sentences have been replaced by other. In addition, we add topics summarizing the main results observed.

Round 2

Reviewer 2 Report

Thank you for incorporating the recommended changes to improve its  readability.

Best Wishes, 

Author Response

Thank you for your considerations. All responses are in the document bellow,  please see the attachment. 

Best wishes. 

Reviewer 3 Report

The authors have extensively responded to my comments (and those of the other reviewers). I thank the authors for their improvements, which are substantial, and in my view the manuscript should be made available to the scientific community.

Author Response

Thank you for your considerations, the anwsers are in the file bellow, please see the attachment. 

Best wishes. 
